# ARE LARGE LANGUAGE MODELS GOOD XAI ANNOTATORS?

## ABSTRACT

Explainable AI (XAI) methods for deep neural networks (DNNs) typically rely on costly annotations to supervise concept-class relationships. To alleviate this burden, recent studies have leveraged large language models (LLMs) and vision-language models (VLMs) to automatically generate these annotations. However, the sufficiency of such automated annotations—whether the generated concepts sufficiently characterize their corresponding classes—remains underexplored. In this paper, we propose the *Fast and Slow Effect* (FSE), a unified evaluation framework designed to assess annotation sufficiency without human supervision. FSE first guides the LLMs to progressively annotate concept-class test cases along a continuum, ranging from a *fast mode*, involving opaque visual labeling without any conceptual reasoning, to a *slow mode*, employing a multi-step, conceptual coarse-to-fine annotation strategy. Then, to systematically validate the sufficiency at each step, our framework leverages the models to self-evaluate annotations using the *Class Representation Index* (CRI), a metric designed to measure how sufficiently annotated concepts represent the target classes against semantically similar alternatives. Our experiments reveal that the current annotation methods fail to provide sufficient semantic coverage for accurate concept-class mapping, especially in fine-grained datasets. Specifically, a significant performance gap is observed between fast and slow modes, with the CRI dropping by over 25% on average in slow mode, indicating while the annotators' intrinsic knowledge enables rapid inference, it remains challenging for them to conceptualize this knowledge in the slow mode, making such expertise difficult to access and interpret. These findings underscore the need for more transparent frameworks to enable reliable, concept-aware annotation in XAI.

## 1 INTRODUCTION

Deep neural networks (DNNs) have achieved remarkable success in computer vision tasks (Deng et al., 2009; He et al., 2016), but their complexity limits interpretability, which is crucial in domains such as medical imaging and engineering inspection. Explainable AI (XAI) methods, such as concept-based models (Koh et al., 2020; Oikarinen et al., 2023; Yang et al., 2023; Sun et al., 2024; Srivastava et al., 2024; Radford et al., 2021; Achiam et al., 2023; Wang et al., 2024; Grattafiori et al., 2024), encode human-interpretable concepts to enhance transparency. However, these models require extensive manual annotations from domain experts to supervise concept-class relationships during training, making them costly and difficult to scale (Snow et al., 2008). Recent work has turned to LLMs and VLMs for automated concept annotation. Two distinct annotation paradigms have emerged: (1) *post-hoc annotation*, employing LLMs via textual prompting to generate class-level concept annotations (Oikarinen et al., 2023; Koh et al., 2020; Yuksekgonul et al., 2022; Yang et al., 2023; Sun et al., 2024; Srivastava et al., 2024); and (2) *visual grounded annotation*, utilizing VLMs to directly link visual inputs to concepts and subsequently to classes at the image level (He et al., 2025; Hossain et al., 2024; Patrício et al., 2025; Selvaraj et al., 2024). While such generated annotations are often intuitively plausible, it remains unclear whether they provide *sufficient* semantic coverage to capture the defining essence of their target classes, especially in specialized domains. Without this sufficiency, concept-based explanations risk reflecting superficial correlations rather than genuine discriminative cues (Figure 1).

Figure 1: Motivating example: An automated annotator generates a set of concepts for an image and correctly identifies its class. However, when asked to choose between the correct class and several semantically similar alternatives using only its own concepts, it fails. Despite the initial correct inference, the annotator fails in the second stage, raising concerns about the annotator's true understanding of conceptual relationships and motivating further investigation.

Motivated by these limitations, we propose the *Fast and Slow Effect* (FSE) framework to systematically evaluate conceptual annotation sufficiency without human supervision (see Figure 2). Our framework comprises two main components: (a) annotating test cases for concept–class relations by simulating how existing annotators progressively refine concepts for a target class, which is then structured into five refinement stages—*Background*, *Superclass*, *Salient Features*, *Detailed Features*, and *Auxiliary Features*. In this process, class predictions transit from a *fast mode* (opaque visual inference without any conceptual reasoning) to a *slow mode* (leveraging the accumulated concepts); and (b) *Class Representation Index* (CRI), an evaluation metric, which quantifies how sufficiently the accumulated concepts support accurate concept-class mapping. We further hypothesize a phenomenon termed *Slow Mode Superiority*, where class mapping guided by accumulated concepts will yield higher CRI scores compared to opaque visual inference. This highlights the significance of concept-based textual supervision in enhancing the sufficiency of concept-class relationships.

However, empirical results reveal that the current annotation methods fail to provide sufficient semantic coverage for accurate concept-class mapping, especially in fine-grained datasets. The slow mode significantly reduces performance—by over 25% on average—compared to the fast mode, indicating while the annotators' intrinsic knowledge enables rapid inference, it remains challenging for them to conceptualize this knowledge in the slow mode, rendering such expertise opaque. We further apply our FSE framework to examine the widely adopted *utility-as-proxy* assumption (Hu et al., 2024b;a; He et al., 2025), which posits that if concept knowledge is incorporated into the visual pipeline—enabling joint multi-modal prediction—then improved performance on downstream tasks reflects annotation quality. Surprisingly, our fused mode—which integrates fast and slow modes to simulate such an end-to-end pipeline—achieves an CRI score of approximately 90%, whereas the slow mode alone scores only about 50% under identical conditions. This discrepancy indicates that strong performance in downstream tasks may not necessarily correlate with adequate conceptual supervision, suggesting that high utility scores can be misleading if the annotations are insufficient.

Our key contributions are:

- We propose the *Fast and Slow Effect* (FSE), a fully autonomous framework for validating the sufficiency of automated concept-class annotations without human supervision.

- We propose a novel evaluation metric, the *Class Representation Index* (CRI), designed to quantitatively measure how sufficiently the accumulated conceptual annotations support accurate concept-class mapping, providing interpretable criteria for assessing whether annotations capture sufficient semantic relationships or merely reflect superficial correlations.

- We conduct extensive experiments across diverse and fine-grained datasets, demonstrating that current automated annotators often fail to achieve adequate semantic coverage, underscoring the need for more robust and semantically expressive annotation strategies.

## 2 BACKGROUND

**Concept-based Models.** Concept-based models have emerged as a promising paradigm for enhancing the interpretability of deep neural networks (DNNs) by explicitly incorporating human-understandable concepts into the decision-making process. Notable approaches in this domain include Concept Bottleneck Models (CBMs) (Koh et al., 2020; Oikarinen et al., 2023; Yang et al., 2023; Sun et al., 2024; Srivastava et al., 2024), Contrastive Language-Image Pre-training (CLIP) (Radford et al., 2021), and Vision-Language Models (VLMs) (Achiam et al., 2023; Wang et al., 2024; Grattafiori et al., 2024). These methods typically leverage visual and textual modalities jointly to perform class predictions. Formally, we define a training dataset as $\mathcal{D} = \{(x_i, c_i, y_i)\}_{i=1}^N$, where each data point consists of an input image $x_i \in \mathbb{R}^d$ (with $d$ pixels), a set of concept embeddings $c_i = \{c_{i,1}, c_{i,2}, \ldots, c_{i,M_i}\}$, and a class label $y_i \in \{1, \ldots, K\}$. Here, $K$ denotes the total number of distinct classes, and each concept embedding $c_{i,j} \in \mathbb{R}^{d_c}$ corresponds to a human-understandable textual description associated with the image $x_i$. The dimensionality of the concept embeddings is represented by $d_c$, while $M_i$ indicates the number of concepts annotated for the $i$-th image. The objective is to learn a visual encoder $f_v : \mathbb{R}^d \to \mathbb{R}^{d_z}$ that maps input images to visual features of dimensionality $d_z$, a concept mapping $f_c : \mathbb{R}^{d_z} \to \mathbb{R}^{d_c}$ that projects visual features into the conceptual embedding space, and a prediction head $f_p : \mathbb{R}^{d_c} \to \mathbb{R}^K$ that produces the final class prediction. Despite their promise, a significant bottleneck in deploying concept-based models is the requirement for explicit concept supervision ($c_i$) during training. Acquiring such supervision typically involves manual annotation, which is labor-intensive and challenging to scale to large datasets (Snow et al., 2008). This challenge has spurred significant interest in developing automated methods for generating concept annotations at scale.

**Automated Annotation.** These methods aim to augment datasets that initially contain only class labels with explicit concept annotations. Formally, given a class labeled dataset without concept annotations: $\mathcal{D}_{\text{cls}} = \{(x_i, y_i)\}_{i=1}^N$, where each input image $x_i \in \mathbb{R}^d$ is associated only with a class label $y_i \in \{y_k\}_{k=1}^K$, the objective is to automatically generate a set of concepts $c_i$. Automated annotation methods can be broadly divided into two main categories: *post-hoc textual annotation* and *visual-grounded annotation*. Post-hoc textual annotation typically generates domain-specific textual annotations at the class level. Early approaches utilized general-purpose knowledge graphs, such as ConceptNet (Liu & Singh, 2004; Yuksekgonul et al., 2022), to infer structured relationships between concepts and class labels. More recently, LLMs have been employed to generate domain-specific textual concepts (Oikarinen et al., 2023; Yang et al., 2023; Sun et al., 2024; Srivastava et al., 2024). In this setting, annotators are prompted to produce a set of textual concepts $c_k$ describing each class $y_k$, resulting in annotations of the form: $\mathcal{D}_{\text{post-hoc}} = \{(c_k, y_k)\}_{k=1}^K$. While post-hoc annotations improve domain relevance, their abstract, class-level nature often lacks explicit grounding in visual evidence, which can limit both interpretability and precision. In contrast, visual-grounded annotation directly leverages VLMs to generate fine-grained, image-specific concept annotations (He et al., 2025; Hossain et al., 2024; Patrício et al., 2025; Selvaraj et al., 2024). Here, VLMs produce visual-grounded concept annotations $c_i$ for each individual image $x_i$, resulting directly in the training dataset $\mathcal{D}$. By explicitly grounding concepts in visual evidence, the annotation methods enhance interpretability, reduce ambiguity, and provide more precise annotations for downstream modeling tasks.

## 3 MOTIVATION

**Current Limitations in Validating Annotations.** Despite the progress in automated annotation, systematic validation of the generated concept annotations remains an underexplored area. While initial efforts have been made, current validation strategies are primarily confined to two main approaches: *human evaluation* (Oikarinen et al., 2023; Yang et al., 2023; Sun et al., 2024; He et al., 2025) and an approach here referred to as the *utility-as-proxy assumption* (Hu et al., 2024b;a; He et al., 2025). Human evaluation, while intuitive, is fraught with practical and methodological challenges. Recent studies (Ford & Keane, 2022) have shown that human perceptions of explanations can vary significantly depending on domain expertise, affecting response times, perceived helpfulness, and trustworthiness. Moreover, obtaining consistent, high-quality human annotations is inherently difficult, expensive, and infeasible at scale (Snow et al., 2008). The utility-as-proxy assumption, on the other hand, describes a practice common in prior work: assessing the validity of generated concept annotations by measuring their effect on downstream classification accuracy. Although

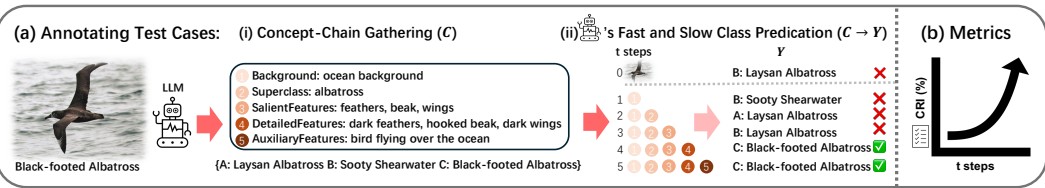

Figure 2: Overview of the proposed FSE framework. The framework consists of two main components: (a) *Annotating Test Cases* for concept-class relations, and (b) *Evaluation Metrics*. In (a), test cases are annotated through an incremental two-step process: (i) concepts are progressively collected over $t$ steps to refine understanding, and (ii) the model maps these concepts to class labels at each step. This process begins with a *fast mode* ($t = 0$), where class labels are directly inferred from visual input without conceptual or textual cues, and transitions to a *slow mode* ($t > 0$), where predictions leverage the accumulated concept set. In (b), we introduce the *Class Representation Index* (CRI) to quantify the likelihood that the accumulated concepts sufficiently represent the target classes against semantically similar alternatives. As annotation steps increase, we expect the CRI to rise.

straightforward, this approach introduces considerable uncertainty. Recent work has demonstrated that end-to-end utility can improve even when annotations are irrelevant or encode unintended shortcuts (Havasi et al., 2022; Sun et al., 2024). Furthermore, as illustrated in the motivating example (Figure 1), annotations that appear beneficial in multimodal fusion may become misleading or insufficient when evaluated in isolation. Consequently, improvements in end-to-end accuracy alone do not reliably reflect the interpretability or sufficiency of the underlying annotations.

**Towards Rigorous Criteria for Annotation Sufficiency.**   Given these limitations, it is essential to develop an evaluation framework that is automatic, requiring no human supervision, and capable of assessing annotation quality beyond mere improvements in downstream accuracy. A critical aspect of this framework is the need to clarify what constitutes a sufficient annotation. Recent advances in LLM research have demonstrated promising self-assessment capabilities, enabling models to critically evaluate their own outputs (Kiciman et al., 2023; Xie et al., 2023; Panickssery et al., 2024). Intuitively, a trustworthy annotation should be self-contained, meaning it must provide all necessary information to sufficiently infer the target class without needing any external context or supplementary information. Motivated by this intuition, we formally define the notion of *sufficient* concept-class annotation as follows:

**Definition 3.1** (Sufficient Concept-Class Annotation). A concept-class annotation generated by an LLM or VLM is considered *sufficient* if the generated concepts alone are expressive, clear, and precise enough to enable accurate inference of the corresponding class, without requiring additional external information or contextual cues.

This definition provides a principled foundation for the development of automatic evaluation frameworks. Building on this, we introduce our proposed approach, the *Fast and Slow Effect* (FSE) framework, which serves as a novel evaluation paradigm for assessing annotation sufficiency, with the following section introducing its specific procedures and metrics.

## 4 FAST AND SLOW EFFECT (FSE) FRAMEWORK

The FSE framework (Figure 2) consists of two main components: (1) annotating test cases for concept-class relations, where concepts are incrementally collected following established concept-gathering paradigms Yuksekgonul et al. (2022); Yang et al. (2023); Panousis et al. (2025); Oikarinen et al. (2023); Sun et al. (2024) to refine conceptual understanding; this ensures our approach is grounded in and extends mainstream hierarchical extraction practices; and (2) an evaluation metric, the proposed *Class Representation Index* (CRI), measuring how sufficiently concepts support accurate concept-class mapping.

### 4.1 ANNOTATING TEST CASES FOR CONCEPT-CLASS RELATIONS

To replicate the concept-based annotation paradigm and investigate concept-class relationships ($C \rightarrow Y$), we construct test cases $\mathcal{D}_{\text{test}}$ of annotated concept-class pairs. This involves multiple

annotation steps to explore these relationships hierarchically from coarse to fine. The test cases are defined as:

$$\mathcal{D}_{\text{test}} = \{(c_i^t, y_i^t) \mid t = 1, \ldots, T; \, i = 1, \ldots, l\},$$

where $c_i^t$ denotes the concepts for instance $i$ at step $t$, and $y_i^t$ is the class mapped post-concept gathering, with $l$ as the total number of cases. The annotation proceeds in two stages: (i) *Concept-Chain Gathering*, where concepts are incrementally refined, and (ii) *Fast and Slow Class Prediction*, mapping concepts to their corresponding classes after each gathering step.

**Concept-Chain Gathering.**    Given an input query $X_i$ from a labeled instance $(x_i, y_i) \in \mathcal{D}_{\text{cls}}$, we consider two annotation scenarios: *post-hoc* annotation at the class level ($X_i = y_i$, covering all $K$ classes) and *visual-grounded* annotation at the image instance level ($X_i = x_i$, covering all $N$ samples). We then initiate a five-stage annotation process ($T = 5$) that progressively refines concepts from Stage 1 to Stage 5 for each $X_i$. The choice of five stages *reflects* and builds upon established methodologies for structured, hierarchical concept extraction, which typically progress from coarse to fine levels of detail. For example, certain methods(Yuksekgonul et al., 2022; Yang et al., 2023) use a single-level process that directly produces concepts without further hierarchical refinement. Sun et al. (2024); Panousis et al. (2024) adopt a two-level scheme (*Perceptual* vs. *Descriptive*), while Oikarinen et al. (2023) propose a three-tier process (*Background*, *Superclass*, *Important Features*).

We extend these ideas into the following **five-stage refinement process**:

1. *Background* – High-level environmental or contextual cues.
2. *Superclass* – Broad categorical grouping of the object.
3. *Salient Features* – Prominent visual traits that are visually distinctive.
4. *Detailed Features* – Fine-grained and discriminative characteristics per salient feature.
5. *Auxiliary Features* – Supplemental attributes to enhance coverage and completeness.

The refinement begins with coarse concepts such as *"Background: Ocean"* or *"Superclass: Bird"*, and gradually incorporates finer attributes, e.g., *"Narrow and pointed wings"*. Formally, the concept chain at step $t$, denoted $c_i^t$, is obtained from the annotator model $\mathcal{F}$ as:

$$c_i^t = \bigcup_{j=1}^{t-1} \mathcal{F}(c_i^j, X_i; \Theta), \quad t = 1, \ldots, T, \tag{1}$$

where $\mathcal{F}$ is a fixed LLM/VLM-based annotator that refines the concept set based on the previous output $c_i^{t-1}$ and the query $X_i$. The parameters $\Theta$ capture the annotator's model weights.

**Fast and Slow Class Prediction.**    Immediately after each concept-gathering step $t$, the model synthesizes the accumulated concept set $c_i^t$ into a class prediction $y_i^t$. To systematically investigate contradictions between raw visual inputs and conceptual annotations (illustrated in Figure 1), we categorize predictions into two distinct modes based on the annotation step $t$:

- *Fast Mode* ($t = 0$): In this mode, classes are annotated directly from the visual input $x_i$ without intermediate textual annotations:

$$y_i^0 = \mathcal{F}(x_i; \Theta).$$

  This mode applies exclusively to visual-grounded scenarios, where the input $X$ visually represents the class $y$. The post-hoc scenario inherently requires explicit conceptual annotations and thus is not suitable for this mode.

- *Slow Mode* ($t > 0$): In contrast, the slow mode is applicable to both visual-grounded and post-hoc scenarios, where predictions involve a structured, multi-step textual annotation process, incrementally gathering and refining conceptual information before each prediction. Importantly, at these stages, the original input $X_i$ is no longer required, and the prediction relies solely on the high-level conceptual annotations:

$$y_i^t = \mathcal{F}(c_i^t; \Theta), \quad t = 1, \ldots, T.$$

### 4.1.1 PROMPT DESIGN

We employ the structured hierarchical prompting strategy previously described, comprising $T = 5$ concept-gathering stages. Detailed prompt formulations for each stage are provided in Appendix B.

After each concept-gathering step $t$, the model uses only the textual concepts collected up to that step to predict $y_i^t$ from a candidate set $S = \{y_i, d_i^j\}_{j=1}^4$, which contains the ground-truth class $y_i$ and four semantically similar distractor classes $d_i^j$. We carefully construct this candidate set to realistically challenge the model, as detailed in Section 5.3.

## 4.2 METRICS

**Class Representation Index (CRI).** Given the set of annotated test cases $\mathcal{D}_{\text{test}}$ and their corresponding class labeled dataset $\mathcal{D}_{\text{cls}}$, the CRI quantifies the likelihood that the concept information alone supports accurate classification, *e.g.*, the proportion of correctly predicted labels $y_i^t$ compared to the ground-truth labels $y_i$ from $\mathcal{D}_{\text{cls}}$. Formally, the CRI at step $t$ is defined as:

$$CRI(\mathcal{F}, t; \mathcal{D}_{\text{test}}, \mathcal{D}_{\text{cls}}) := 100\% \times \frac{1}{l} \sum_{i=1}^{l} \mathbb{1}\big[y_i^t = y_i\big], \tag{2}$$

$$\text{where} \quad y_i^t = \begin{cases} \mathcal{F}(x_i; \Theta), & \text{if } t = 0 \\ \mathcal{F}(c_i^t; \Theta), & \text{if } t > 0 \end{cases}$$

We will often write it as $CRI(t)$ or just $CRI$ to simplify notation. A higher CRI indicates that the conceptual annotations at step $t$ provide a sufficient foundation for classification. A well-structured concept chain should exhibit positive incremental CRI at each annotation step. Specifically, a positive marginal CRI increment ($CRI(t) - CRI(t-1) > 0$) indicates that the annotation at step $t$ provides valid conceptual information, whereas a non-positive increment suggests insufficiency at that step.

**Slow Mode Superiority.** According to the dual-process theory (Kahneman, 2011), fast mode serves as a "black box" approach, relying on direct visual conclusions without extensive reasoning, which can lead to quick but less thoughtful results. Slow mode, on the other hand, involves a detailed, conceptual, and multi-step reasoning process, which is more thorough but time-consuming. Therefore, when both modes are available, the slow mode is expected to consistently achieve performance superior or at least comparable to the fast mode. Specifically, we consider the slow mode at its maximum annotation step $t = T$, representing the scenario where the annotator has fully leveraged all available annotation opportunities. Formally, we define CRI-Gap $\Delta CRI_T$ between the slow mode (at step $t = T$) and the fast mode (at step $t = 0$) as:

$$\Delta CRI_T = CRI(T) - CRI(0). \tag{3}$$

We expect this CRI gap to be non-negative, indicating the superiority (or at least equivalence) of the slow mode: $\Delta CRI_T \geq 0$.

## 5 IMPLEMENTATION

### 5.1 DATASET

Following previous works (Oikarinen et al., 2023; Koh et al., 2020; Yuksekgonul et al., 2022; Yang et al., 2023; Sun et al., 2024; Srivastava et al., 2024), we evaluate our framework on three fine-grained visual classification datasets (*CUB-200 Birds* (Welinder et al., 2010), *Cars-196* (Krause et al., 2013), *Flowers-102* (Nilsback & Zisserman, 2008)) and two general object recognition datasets (*CIFAR-100* (Krizhevsky et al., 2009), *Caltech-101* (Li et al., 2022)). Detailed descriptions for each dataset are provided in Appendix A.

### 5.2 MODEL SELECTION

To effectively evaluate the annotation performance of LLMs and VLMs, we select several representative models reflecting recent advancements in textual reasoning and visual understanding. Given rapid developments in multimodal capabilities, we prioritized models capable of both purely textual (post-hoc) and visual-grounded annotation tasks, ensuring fairness and consistency in our evaluation. We chose three prominent model families: GPT-4o (Achiam et al., 2023), Qwen2-VL (Wang et al., 2024), and Llama3.2 (Grattafiori et al., 2024). For a balanced assessment, we evaluated two model

sizes from each family, covering both large-scale and smaller-scale variants: GPT-4o, GPT-4o-mini, Llama-3.2-vision-90b, Llama-3.2-vision-11b, QwenVL2-72b, and QwenVL2-7b. For simplicity, we refer to these multimodal models as LLMs throughout this paper. Additionally, to demonstrate the effectiveness of our FSE framework, it supports evaluation of Chain-of-Thought (CoT) performance for models specifically designed for reasoning tasks. Notably, our application of the FSE revealed that even advanced reasoning models like DeepSeek-R1 (Guo et al., 2025) often bypass their own detailed CoT reasoning processes in decision-making, highlighting limitations in their reasoning abilities. Please refer to Appendix D for further details.

Table 1: Contradiction rates (%) of GPT-series models when predicting $y^{con}$ using generated concepts under two different candidate set construction strategies. A contradiction occurs when the concept-based prediction $y^{con}$ differs from the initial prediction $y^{init}$, indicating inconsistency between the model's initial output and its concept-driven prediction. For evaluation, we use three datasets (Car, Flower, and CUB-Bird), randomly sampling 100 images from each dataset, and report the contradiction rates averaged across these samples.

| Strategy | Model | Car | Flower | CUB-Bird | Average |
|---|---|---|---|---|---|
| Semantically Related Selection | GPT-4o | 42.39 | 14.14 | 45.90 | 34.14 |
| | GPT-4o-mini | 41.30 | 35.35 | 59.02 | 45.22 |
| Random Selection (Baseline) | GPT-4o | 18.48 | 6.06 | 18.03 | 14.19 |
| | GPT-4o-mini | 14.13 | 17.16 | 31.15 | 20.81 |

### 5.3 PRELIMINARY EXPERIMENT: SELECTING EFFECTIVE DISTRACTOR STRATEGIES

Before presenting our main results, we first conduct a preliminary contradiction test to identify the most effective strategy for selecting distractor classes used in our FSE evaluation. This preliminary experiment provides a glimpse into the annotation quality by evaluating how well different distractor strategies challenge the annotators, with the goal of ensuring that the candidate set (S) used in subsequent evaluations realistically challenges the annotators. We consider two candidate distractor selection strategies:

1. *Random Selection:* Distractor classes are randomly chosen from the entire set of available classes, without considering semantic or visual relationships.

2. *Semantically Related Selection:* Distractor classes are selected based on semantic similarity. Specifically, we construct a Semantic Similarity Dictionary (SSD) using predictions from a pretrained ResNet-18 (He et al., 2016). For each class, we record the top four predicted classes (excluding the ground-truth class itself) for each data sample. These top predictions serve as semantically related distractors.

To evaluate these strategies, we simplify the FSE framework into a contradiction test. Given an image sample $x_i$, we prompt GPT-4 annotators to generate an initial prediction ($y_i^{init}$) as well as their related descriptive concepts ($C_i$), as shown in Figure 1. Next, annotators must select the correct class from the candidate set $\{y_i^{init}, d_i^j\}_{j=1}^4$ using only the generated concepts $C_i$, producing a second prediction ($y_i^{con}$). Here, each $d_i^j$ is a distractor class selected according to one of the two strategies described above. A contradiction occurs when the annotator's initial prediction differs from the second prediction ($y_i^{init} \neq y_i^{con}$), indicating that the distractors effectively challenge the annotator's reasoning. To avoid positional bias (Shi et al., 2024), we randomly shuffle the candidate set in both strategies. Table 1 summarizes the results of this experiment. We observe that random selection yields relatively low contradiction rates (14–20%), suggesting that randomly chosen distractors are ineffective at challenging annotators. In contrast, semantically related selection significantly increases contradiction rates (34–45%), demonstrating its effectiveness in creating challenging candidate sets. Based on these findings, we adopt the Semantically Related Selection for all subsequent experiments.

## 6 RESULTS

**CRI Comparison.** Figure 3 summarizes the CRI scores achieved by six representative LLMs across three specialized fine-grained datasets. In the post-hoc textual annotation scenario, the CRI

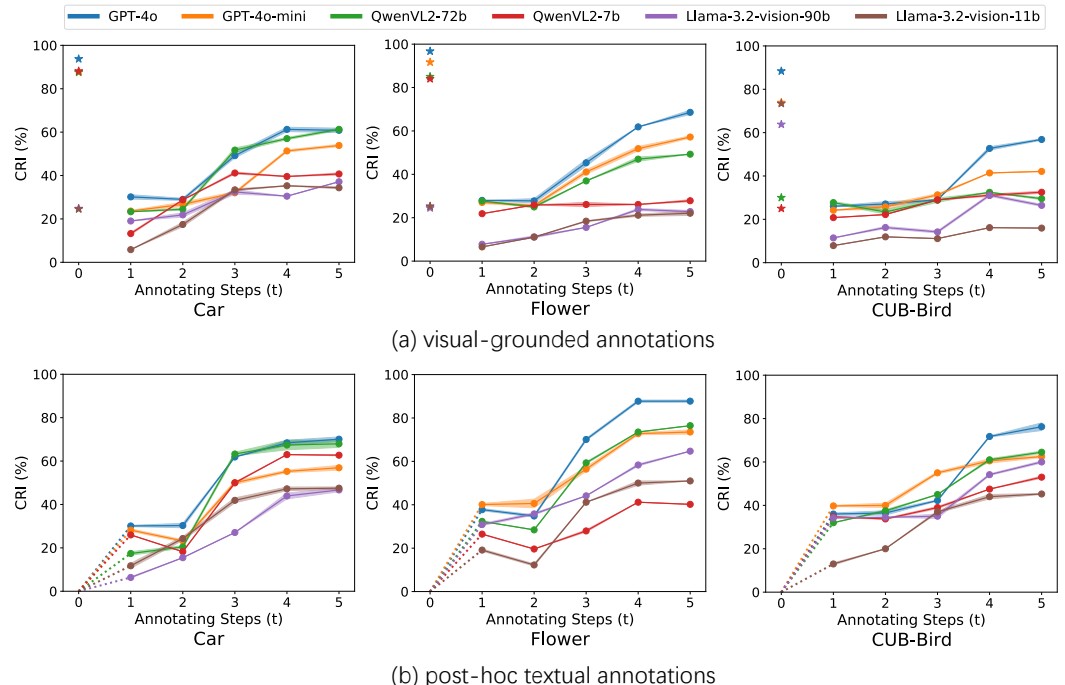

(a) visual-grounded annotations

(b) post-hoc textual annotations

Figure 3: CRI (%) of LLMs Across Annotation Steps as an Indicator of Annotation Sufficiency. (a) shows results for the visual-grounded annotation scenario, and (b) shows results for the post-hoc textual scenario. The dot marker ($\cdot$) denotes the slow mode ($t > 0$) in both scenarios, while the star marker ($\star$) denotes the fast mode ($t = 0$), which is only applicable in the visual-grounded scenario (a). For each data point, three runs (with different seeds) were conducted, and the shaded regions represent the error bars (e.g., standard deviation), it is clear that the standard deviations are negligible, indicating that the results are consistent across repeated trials, and that the observed trends are not due to random variation. Annotation sufficiency generally improves, though the magnitude of improvement varies.

Table 2: Comparison of CRI-Gap (%) between the slow mode at maximum annotation steps ($t = 5$) and the fast mode ($t = 0$), calculated using Eq. 3. Positive values indicate better annotation sufficiency in the slow mode, while negative values suggest the opposite.

| Dataset | GPT-4o | GPT-4o-mini | Llama-3.2-vision-90b | Llama-3.2-vision-11b | QwenVL2-72b | QwenVL2-7b | Average |
|---------|--------|-------------|----------------------|----------------------|-------------|------------|---------|
| Car | -32.92 | -33.65 | -12.36 | -9.78 | -26.43 | -47.43 | -27.10 |
| Flower | -28.19 | -34.50 | -1.66 | 3.24 | -35.74 | -56.18 | -25.51 |
| CUB-Bird | -31.56 | -31.79 | -37.36 | -57.44 | -0.46 | 7.50 | -25.19 |

scores for the Car and CUB-Bird datasets generally remain below 70%, with only the Flower dataset occasionally surpassing 80%. The visual-grounded annotation scenario proves even more challenging, as all models achieve CRI scores below 60% even when the annotator fully leverages all available conceptual annotation opportunities (e.g., $t = 5$). These results highlight the persistent limitations of current LLM-generated annotations in addressing complex, fine-grained classification tasks. We further explore whether the slow mode offers advantages over the fast mode. Table 2 presents the CRI score differences. Contrary to initial expectations, the slow mode frequently underperforms compared to the fast mode on specialized datasets, with average CRI gaps ranging from $-25\%$ to $-27\%$. This finding suggests that while the annotators' intrinsic knowledge enables rapid inference, it remains challenging for them to conceptualize this knowledge in the slow mode. Even when the LLMs are guided through a concept-chain process consisting of five distinct stages intended to make their annotation explicit, the models still struggle to externalize their implicit expertise. As a result, much of their expertise remains opaque and difficult to leverage for downstream knowledge transfer.

**Results on Common Datasets.** To assess whether these limitations are pervasive across datasets, we extend our analysis to common object recognition datasets (CIFAR-100 (Krizhevsky et al., 2009)

and Caltech-101 (Li et al., 2022)) using GPT-4o and GPT-4o-mini in visual-grounded scenarios (Table 3). Remarkably, we observe a completely opposite trend in this context. Both models achieve high CRI scores exceeding 90% at $t = 5$, representing a substantial improvement over their performance on specialized datasets. Furthermore, for the first time, we observe that the slow mode consistently outperforms the fast mode on these general datasets. This indicates that LLMs are capable of generating discriminative and sufficient concept sets when the annotation task is less fine-grained and more general in nature.

Table 3: CRI (%) of GPT-4o and GPT-4o-mini across annotation steps ($t$) in visual-grounded scenarios. Results are shown for general object recognition datasets (CIFAR-100 and Caltech-101). "FineGrained-Avg" denotes the average CRI score computed across the three fine-grained datasets presented in Figure 3.

| Model | Dataset | CRI Score (Steps $t$) | | | | | |
| | | 0 (Fast) | 1 | 2 | 3 | 4 | 5 |
|---|---|---|---|---|---|---|---|
| GPT-4o | CIFAR-100 | 84.84 | 29.23 | 64.40 | 83.96 | 91.43 | **94.07** |
| | Caltech-101 | 91.48 | 30.88 | 80.17 | 91.50 | 93.77 | **93.77** |
| | FineGrained-Avg | **92.97** | 27.67 | 27.11 | 40.28 | 58.54 | 61.97 |
| GPT-4o-mini | CIFAR-100 | 83.79 | 33.89 | 67.16 | 84.84 | 90.53 | **95.37** |
| | Caltech-101 | 89.01 | 33.79 | 76.10 | 85.99 | 87.09 | **89.56** |
| | FineGrained-Avg | **84.37** | 25.02 | 25.47 | 34.14 | 48.69 | 51.01 |

**Utility-as-Proxy $\not\Rightarrow$ Annotation Sufficiency.** We further leverage our FSE framework to critically examine the validity of the widely adopted utility-as-proxy evaluation paradigm (Hu et al., 2024b;a; He et al., 2025) for annotation quality. To closely replicate this evaluation scenario, we fuse the fast mode ($t = 0$) and slow mode ($t = 5$) during classification, rigorously simulating the end-to-end inference pipeline commonly employed by standard concept-based multimodal models. Specifically, during prediction, the LLMs jointly receive both the visual image and its corresponding generated textual annotation as inputs to determine the class labels. We report the results for GPT-4o and GPT-4o-mini in Table 4. Notably, the CRI score obtained through this fusion approach closely aligns with that of the fast mode alone and significantly surpasses the performance of the slow mode. This discrepancy indicates that strong performance in downstream tasks may not correlate with adequate conceptual supervision, suggesting that high utility can be misleading if the underlying conceptual annotations are insufficient.

Table 4: CRI (%) among three annotation modes on three specialized datasets.

| Model | Dataset | Mode of annotation | | |
| | | Fast | Slow | Fuse |
|---|---|---|---|---|
| GPT-4o | Car | 93.75 | 60.82 | 93.08 |
| | Flower | 96.76 | 68.57 | 96.14 |
| | CUB-Bird | 88.40 | 56.84 | 83.60 |
| GPT-4o-mini | Car | 87.50 | 53.85 | 85.75 |
| | Flower | 91.70 | 57.19 | 83.60 |
| | CUB-Bird | 73.90 | 42.11 | 65.80 |

**Visual Case Study.** We also provide a detailed visual analysis to further illustrate the limitations and insufficiency of current LLM-generated annotations. Please refer to Appendix C for specific visual examples highlighting scenarios where LLM-generated annotations fall short.

## 7 CONCLUSION

In this paper, we present the FSE evaluation framework to assess the sufficiency of concept-class annotations in XAI methods. Our extensive experiments shed light on the shortcomings of current annotation methods, revealing that they often fail to adequately capture class semantics, particularly in fine-grained datasets. We encourage future work to leverage our findings to create more effective annotation strategies that improve XAI quality and interpretability.

## 8 ETHICS AND LIMITATIONS

We propose the FSE evaluation framework to assess the sufficiency of concept-class annotations in XAI methods. Our aim is to advocate for a more transparent and concept-aware annotation framework, which has the potential to significantly enhance the interpretability and reliability of XAI systems. By illuminating the challenges that annotators encounter in slow, knowledge-intensive tasks, this work can inform the development of future tools and methodologies that foster improved human-AI collaboration, particularly in domains that require high levels of trust and interpretability. However, in terms of ethical considerations, it is important to acknowledge the potential negative societal impacts associated with the FSE framework. Its reliance on controlled, open-sourced datasets may not fully capture the complexities of real-world data, which could lead to biased or incomplete annotations. This is especially concerning in sensitive sectors such as healthcare, finance, and criminal justice, where such biases could inadvertently contribute to inequalities in decision-making. We are committed to continuously refining and enhancing the framework to address these challenges and ensure its broader applicability in real-world contexts.

## 9 REPRODUCIBILITY

We have provided the code and data at here.

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

# APPENDIX

## A    DATASET DETAILS

We follow previous works (Oikarinen et al., 2023; Koh et al., 2020; Yuksekgonul et al., 2022; Yang et al., 2023; Sun et al., 2024; Srivastava et al., 2024) in selecting three fine-grained visual classification datasets and two general object recognition datasets for evaluation.

**Fine-grained datasets:** *CUB-200 Birds* (Welinder et al., 2010) contains 11,788 images of 200 bird species, exhibiting high intra-class variation in plumage, pose, and background. *Cars-196* (Krause et al., 2013) comprises 16,185 images of 196 distinct car models, spanning different manufacturers and years, requiring attention to fine differences in shape and design. *Flowers-102* (Nilsback & Zisserman, 2008) consists of 8,189 images of 102 flower species, with substantial diversity in color, petal arrangement, and scale.

**General object recognition datasets:** *CIFAR-100* (Krizhevsky et al., 2009) contains 60,000 low-resolution images ($32 \times 32$ pixels) across 100 everyday object categories. *Caltech-101* (Li et al., 2022) includes 9,146 images covering 101 object categories, including animals, vehicles, and household items, with moderate resolution and varied backgrounds.

## B    PROMPT DESIGN

**Concept-Chain Gathering.** When querying the LLM annotators, we use the following prompt template:

**Prompt:** Based on the provided [entity], please adhere to a systematic approach, progressing from coarse concepts to finer details, to "step-by-step" generate the complete set of concepts associated with [entity].

**Background:** Provide a brief description of the overall background in which the object exists or is used, including its typical environment, purpose, and user base, such as 'ocean background', 'urban setting', or 'beach scenery'.

**Superclass:** Identify the general superclass of the entity, such as 'albatross bird' or 'saloon car'.

**SalientFeatures:** List distinctive features or attributes that make it recognizable or unique.

**DetailedFeatures:** Offer a detailed description of each feature within the entity, including attributes like shape, color, size, and other distinctive characteristics. For example, features might be detailed as 'a red beak' or 'a spoked wheel'.

**AuxiliaryFeatures:** Document any supplementary characteristics, secondary functionalities, or additional attributes not previously mentioned.

The Concept-Chain Gathering process follows a hierarchical, coarse-to-fine strategy. Specifically, the conceptual space is systematically explored by progressively refining broad, general concepts into increasingly detailed and precise attributes. To naturally reflect this hierarchical refinement, we structure the prompt into five intuitive steps, starting from general contextual information (e.g., background and superclass) and gradually progressing toward detailed and specific attributes. Additionally, the final auxiliary features step is included to capture supplementary characteristics and secondary functionalities, ensuring the completeness and comprehensiveness of the resulting concept-chain gathering. This structured approach ensures clarity, reduces ambiguity, and enhances the precision of the final conceptual representation. The placeholder "[entity]" in the prompt is designed to accommodate both visual-grounded inputs (images) and post-hoc textual class queries, making the prompt versatile for different querying scenarios.

**Fast and Slow Class Prediction.** Immediately after each concept-gathering step $t$, the model synthesizes the accumulated concept set $c_i^t$ into a class prediction $y_i^t$.

*Fast Mode.* In this mode, classes are annotated directly from the visual input without intermediate textual annotations. The provided multiple-choice format explicitly forms the *selection set*, consisting of one correct class and four random selected distractor classes. When constructing the selection set, we adopt the *Semantically Related Selection* strategy (as detailed in Section 5.3), as this approach more accurately reflects the model's genuine capability to differentiate the correct class from semantically similar alternatives.

**Prompt:** What species is this? Answer directly with only the option's letter from the given choices (A, B, C, D, or E), without any explanations:

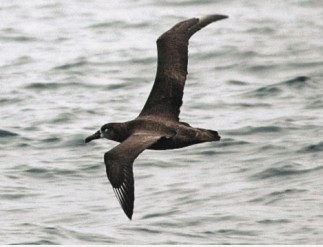

A. [CLS A]    B. [CLS B]    C. [CLS C]    D. [CLS D]    E. [CLS E]

*Slow Mode.* Here, the original input $X$ (image or textual class query) is no longer directly utilized. Instead, the model relies exclusively on the generated textual conceptual representation. In this prompt, the placeholder `HierarchicalConceptJSON` is constructed by selecting reasoning steps up to a specified depth $t$ ($1 \leq t \leq 5$) from the previously generated *Concept-Chain Gathering*, to evaluate the CRI score (Definition 3.1). By systematically varying the annotating depth $t$, we can

quantitatively assess how different levels of conceptual granularity—from coarse concepts at lower levels to finer-grained details at higher levels—impact the alignment between the model's predicted representation of concept-to-class relations and the actual conceptual relations.

---

**Prompt:**

Given the hierarchical conceptual representation generated from the previous reasoning steps (provided as [`HierarchicalConceptJSON`]), identify the correct class label for the described entity. Your answer must strictly be the letter corresponding to the correct class from the following selection set, answer directly with only the letter (A, B, C, D, or E):

A. [CLS A]    B. [CLS B]    C. [CLS C]    D. [CLS D]    E. [CLS E]

---

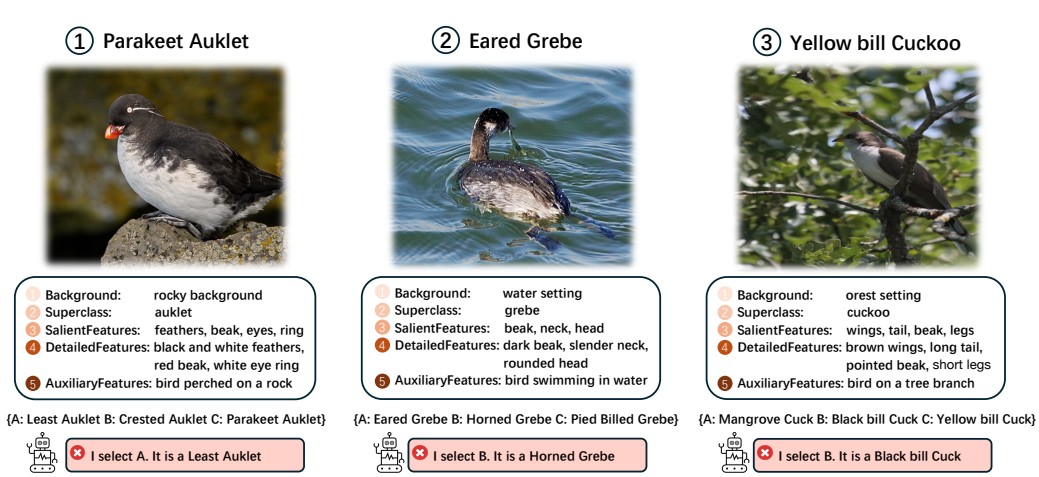

Figure 4: Examples of GPT-4o-generated annotations illustrating limitations in fine-grained bird species annotation. Each subfigure shows a case where GPT-4o correctly identifies the superclass but fails to distinguish between visually similar species due to missing subtle yet critical visual details. ① Parakeet Auklet misclassified as Least Auklet; ② Horned Grebe misclassified as Eared Grebe; ③ Yellow-billed Cuckoo misclassified as Black-billed Cuckoo.

## C    VISUAL CASE STUDY

In this section, we analyze three representative annotation examples generated by GPT-4o from the CUB-Bird dataset (see Figure 4). We specifically select GPT-4o annotations for this analysis because GPT-4o comprehensively achieves the highest CRI score at the maximum annotation steps, as demonstrated in Figure 3. These examples illustrate situations where the annotations, although generally accurate at the superclass level, lack sufficient detail to reliably distinguish between visually similar bird species. This observation suggests that fine-grained annotation tasks may require annotations that incorporate more specialized domain knowledge and subtle visual distinctions. Figure 4 presents three cases where GPT-4o correctly identified the general superclass (Auklet, Grebe, and Cuckoo, respectively, in ①, ②, and ③), but did not correctly classify the specific species. Upon closer inspection, we observe that the annotations omit certain subtle yet important visual characteristics that are critical for accurate species-level identification.

For example, in ①, the bird shown is a Parakeet Auklet, but it was annotated as a Least Auklet. According to (Kaufman, 2000), the primary distinguishing features between these two species include size and bill shape: Least Auklets are smaller with a short bill, whereas Parakeet Auklets are larger and have a distinctive orange, upward-curved bill. The annotation provided did not include these distinguishing details, making accurate species-level classification challenging. Similarly, in ②, GPT-4o confused a Horned Grebe with an Eared Grebe. The key visual difference between these two species lies in their head plumage: Horned Grebes have golden feather tufts extending straight back from the head, resembling horns, while Eared Grebes have fan-shaped golden feathers spreading

outward around the head. The absence of these subtle visual cues in the annotation likely contributed to the misclassification. Finally, in ③, GPT-4o was unable to differentiate between the Yellow-billed Cuckoo and the Black-billed Cuckoo. The primary distinguishing characteristic between these two species is the bill color, yet the annotation did not explicitly mention this feature. Without this critical detail, distinguishing between these two closely related species becomes difficult.

Overall, these examples highlight the potential need for annotations that incorporate more comprehensive domain-specific knowledge and subtle visual distinctions to further improve fine-grained classification performance.

## D  FSE ANALYSIS FOR REASONING MODELS

To further demonstrate the versatility and effectiveness of our proposed Fast and Slow Effect (FSE) framework, we explore its capability to self-evaluate the reasoning chains generated by advanced reasoning models, such as DeepSeek-R1 (Guo et al., 2025). Specifically, we investigate whether the long Chain-of-Thought (CoT) reasoning produced by these sophisticated models aligns naturally with the intuitive, step-by-step inference paradigm that our FSE framework explicitly encourages.

Recall that our FSE framework consists of two primary stages: the concept gathering stage and the class prediction stage. Both stages are designed to follow a natural and intuitive reasoning paradigm, closely resembling the slow, deliberate, and step-by-step thinking process described in cognitive science literature. Given that the Chain-of-Thought prompting strategy similarly aims to elicit explicit reasoning steps from advanced models, we hypothesize that the reasoning chains generated by models such as DeepSeek-R1 will naturally exhibit a similar structure and granularity to our manually designed prompting strategies.

To test this hypothesis, we adapt our concept gathering procedure for DeepSeek-R1. Instead of explicitly prompting the model with carefully designed step-by-step instructions, we employ a simpler and more general prompt:

> **Prompt:** How to step-by-step classify an object as this [entity]?

From the model's response, we extract only the reasoning portion enclosed within the `<think></think>` tags. This extracted reasoning chain serves as the set of gathered concepts for subsequent analysis.

However, a practical challenge arises: our original concept-gathering strategy explicitly defines five distinct reasoning stages ($1 \leq t \leq 5$), which are subsequently utilized in the CRI evaluation (Section 6). Without explicitly prompting the model to produce exactly five reasoning steps, it is unclear how to segment the naturally generated long CoT into discrete stages.

Interestingly, upon examining the reasoning chains generated by DeepSeek-R1, we observe a consistent and natural segmentation pattern. Specifically, the model spontaneously structures its reasoning into distinct steps, each clearly indicated by the paragraphing symbol '>' within its generated CoT. To quantify this observation, we computed the average number of reasoning steps (indicated by the '>' symbol) across three benchmark datasets. The results, summarized in Table 5, reveal that the average number of reasoning steps naturally produced by DeepSeek-R1 closely aligns with our original design choice of five stages.

Table 5: Average number of reasoning steps (indicated by the '>' symbol) spontaneously generated by DeepSeek-R1 across three benchmark datasets. The results closely match our original FSE design choice of five reasoning stages, highlighting the naturalness and appropriateness of our framework's design.

| Dataset | Avg. Number of Reasoning Steps |
|---------|--------------------------------|
| CUB-Bird | 5.2 |
| FLOWER | 4.7 |
| CAR | 5.7 |

Table 6: CRI (%) of DeepSeek-R1 across annotation steps ($t$) in post-hoc textual scenarios. The symbol "/" indicates scenarios where the corresponding mode is not supported.

| Model | Dataset | CRI (Steps $t$) | | | | | |
|-------|---------|-----------|-------|-------|-------|-------|-------|
| | | 0 (Fast) | 1 | 2 | 3 | 4 | 5 |
| | Car | / | 84.62 | 82.14 | 90 | 98 | 98 |
| GPT-4o | Flower | / | 78.02 | 94 | 92.13 | 86.20 | 92 |
| | CUB-Bird | / | 60.3 | 80 | 86 | 84 | 82.31 |

Given this natural alignment, we adopt a straightforward strategy for the subsequent class prediction stage. Specifically, we evenly partition the extracted reasoning chain into five segments, each representing one reasoning stage. These segments are then directly utilized to construct the `HierarchicalConceptJSON` structure. The remainder of the prompting procedure remains unchanged from our original design.

> First, males are very colorful. They have bright blue heads, red underparts, and green backs. Females and juveniles are more subdued, usually a greenish yellow. So I should describe the plumage differences between males and females without using the name.

> They are small songbirds, about the size of a sparrow. Their bill is short and conical, suited for seeds. Maybe mention the habitat: they breed in shrubby areas and thickets, often in the southeastern US, and winter in similar habitats in Florida, the Caribbean, and Central America.

> Their song is a series of high pitched, melodious notes. Also, they're often seen feeding on seeds and insects. I need to avoid terms like "passerina ciris" or the common name. Focus on physical traits, behavior, habitat, and geographic range. Make sure not to list the name anywhere in the reasoning, just in the final answer.

Okay, the user wants to know how to classify a species as an without directly mentioning its name in the reasoning. Let me start by recalling the key features of this bird. First, the male has bright yellow plumage during breeding season, which becomes duller in winter. Females are more olive yellow. They have a small, conical beak suited for eating seeds. Their wings are black with white bars, and they have a distinctive flight pattern with dips and undulations. They're often found in open fields, weedy areas, and gardens, especially where thistles and sunflowers grow. Their song is a series of musical twitters and calls. Also, they're strict vegetarians, mainly eating seeds. They might be confused with other yellow birds like warblers, but the beak shape and wing patterns are different. Their nesting habits involve building compact cups in shrubs. Considering all these traits together helps in accurate classification.

(a) Painted Bunting          (b) American Goldfinch

Figure 5: Examples illustrating abstraction limitations in the R1 model's long CoT reasoning process on the CUB-Bird dataset. (a) Painted Bunting: The reasoning primarily emphasizes male-female distinctions, neglecting broader species-level abstraction. (b) American Goldfinch: The reasoning is presented as a single paragraph without hierarchical structuring, again overly focusing on gender differences rather than comprehensive species characteristics.

## D.1 RESULTS AND ANALYSIS

We observe from Table 6 that although DeepSeek-R1 consistently achieves high CRI scores, this phenomenon is not necessarily indicative of good model behavior. Instead, it strongly suggests the presence of the *Early-Stage High CRI Effect*, wherein annotators achieve disproportionately high CRI scores during the initial annotation steps. This effect typically arises when the model generates invalid or superficial concept annotations, bypassing structured annotation processes and directly inferring labels from shallow visual or textual cues. For instance, initial annotations often represent background or overly general information, which provides limited insight into the specific classes of interest. Specifically, the model attains an unusually high CRI score (around 60-80%) even at the initial reasoning stage, which is typically unexpected. In a proper step-by-step reasoning process from coarse to fine granularity, the initial stages usually provide general or background-level information, offering limited specificity regarding the target classes. Consequently, achieving such high CRI scores at the early stages implies that the model may be bypassing the intended structured annotation process. Rather than progressively refining its reasoning, the model likely relies on superficial cues to directly infer labels, resulting in annotations that are potentially shallow, invalid, or lacking meaningful conceptual depth.

## D.2 REASONING CASE STUDY: CUB-BIRD DATASET

In the previous section, we observe that DeepSeek-R1 exhibits notably high CRI scores during the early stages of reasoning. We hypothesize that this behavior arises primarily from the model's

limitations in maintaining consistent abstraction and hierarchical organization throughout its reasoning process.

A key issue identified is the insufficient granularity and hierarchical clarity within the generated CoT. Specifically, the model frequently produces reasoning chains that either fail to generalize beyond superficial distinctions or lack a clear hierarchical structure. For example, when reasoning about the Painted Bunting (see Figure 5, example (a)), the model predominantly emphasizes superficial differences between male and female birds. Although these distinctions are relevant, the model neglects to provide a broader, comprehensive characterization of the species as a whole. This narrow focus limits the model's ability to abstract effectively, resulting in reasoning that is overly specific and incomplete.

Similarly, in the case of the American Goldfinch (Figure 5, example (b)), the model presents its reasoning as a single, unstructured paragraph without clear hierarchical indicators (such as the '>' symbol). This lack of structured organization further illustrates the model's difficulty in clearly delineating abstract reasoning layers. As with the Painted Bunting example, the reasoning again disproportionately emphasizes gender-based distinctions rather than offering a balanced, comprehensive abstraction at the species level.

These illustrative examples highlight the necessity for improved abstraction granularity and hierarchical structuring within the reasoning processes of DeepSeek-R1. Addressing these shortcomings would significantly enhance the model's ability to generalize effectively, resulting in more coherent, comprehensive, and robust reasoning outputs.

## E  MORE RESULTS

### E.1  IMAGENET

To complement the results presented in Table 3, we randomly sampled 400 images from ImageNet for evaluation, with the average performance reported in Table 7. The results show that ImageNet behaves similarly to the "Fine-Grained" category in Table 3, with the fast mode outperforming the slow mode. We believe this is because, although ImageNet is considered a general-domain dataset, its 1,000 classes include many with high semantic similarity, making it more like a specialized-domain dataset in practice.

Table 7: CRI (%) of GPT-4o and GPT-4o-mini across annotation steps ($t$) in ImageNet.

| Model | Dataset | CRI Score (Steps $t$) | | | | | |
|---|---|---|---|---|---|---|---|
| | | 0 (Fast) | 1 | 2 | 3 | 4 | 5 |
| GPT-4o | ImageNet | 86.63 | 24.23 | 30.09 | 54.32 | 68.25 | 69.06 |
| GPT-4o-mini | ImageNet | 75.82 | 21.40 | 25.45 | 45.78 | 61.90 | 60.00 |

Table 8: CRI (%) using Top-5 concepts among three annotation modes on three specialized datasets.

| Model | Dataset | Mode of annotation | | |
|---|---|---|---|---|
| | | Fast | Slow | Fuse |
| | Car | 93.75 | 41.38 | 92.82 |
| GPT-4o | Flower | 96.76 | 47.90 | 97.21 |
| | CUB-Bird | 88.40 | 42.56 | 81.52 |
| | Car | 87.50 | 38.30 | 85.00 |
| GPT-4o-mini | Flower | 91.70 | 40.17 | 82.69 |
| | CUB-Bird | 73.90 | 36.89 | 64.26 |

### E.2  PERFORMANCE WITH TOP-5 CONCEPTS

We re-ran the experiment in Table 4 using the five most salient concepts per image, as identified directly by the LLM in its output, with the resulting CRI scores reported in Table 8. We evaluated

three settings: Fast mode: image-only input (same as the original setting) – performance remained unchanged. Slow mode: textual Top-5 concepts only (no image) – performance dropped sharply compared to the original slow mode, reaching near random-guess levels. Fusion mode: Top-5 concepts + image – performance was almost identical to Fast mode and very close to the original fusion setting, despite the prompt instructions explicitly discouraging the use of visual content for reasoning. In practice, the LLM appears to incorporate information from the visual patches into its final decision. These observations strengthen our earlier point: in the fusion setting, classification accuracy is not strongly coupled with the quality of the concepts provided.

