# OpenReview forum: "Are Large Language Models Good XAI Annotators?"
_ICLR.cc/2026/Conference — ICLR 2026 Conference Withdrawn Submission_

### Official Review · Reviewer_N1WD · 2025-10-20

**Soundness:** 2
**Presentation:** 2
**Contribution:** 2
**Rating:** 2
**Confidence:** 4

**Summary:**

This paper evaluates the capabilities of (multimodal) LLMs to produce fine-grained concept annotations. Prior work either relied on human evaluations or used downstream performance as a proxy. The key idea of this paper is that if annotations are sufficient, a LLM should be able to infer the correct class from the annotation alone (Definition 3.1).

The authors investigated concept annotations of (multimodal) LLMs through several stages of annotation refinements (l. 234-240). A so-called fast mode, where the model directly predicts the class from the input image (like in standard object recognition). In the so-called slow mode, the model progressively predicts concept annotations from coarse (e.g., background or superclass) to fine-grained levels (e.g., visual attributes such as the shape or color of bird’s wing).

The experiments show that (multimodal) LLMs generally fail to produce concept annotations that are sufficiently accurate or informative to enable correct class inference from them.

**Strengths:**

* S1: The finding that using more fine-grained annotations (slow mode) hurts the correct class inference for fine-grained data like CUB is surprising and interesting.

* S2: The concept annotation sufficiency definition is quite interesting. It captures well what we want concept annotations to look like.

**Weaknesses:**

* W1: A core issue with the evaluation framework is that a LLM is used to infer the class. There’s no guarantee that this LLM is good in inferring the correct class from the annotations.

* W2: The claim that “strong performance in downstream tasks may not correlate with adequate conceptual supervision” (l. 462/463) is too strong. Also, I’d like to note that these models often weigh the concepts or only use a subset of them on a per-instance basis. Thus, the conclusion seems not warranted. It’d be more meaningful to take, say, the top-5 concepts per image and try to see if models can infer the correct class.

* W3: The finding that (multimodal) LLMs work not that well to capture fine-grained details is well-known. The paper doesn’t provide a novel insight regarding this point.

* W4: The paper’s writing could overall be improved for clarity.

### Comments

* C1: The explanation of “utility-as-proxy assumption” is very unclear and would be good to refine for better clarity.

* C2: Comparing the 0-shot classification performance to the concept annotation is likely not that fair. Because, for example, these models might have seen the dataset, so the fast-mode numbers might be therefore a too optimistic estimate. However, I understand that this might be hard to control for the authors and, thus, only listed it here as a comment.

**Questions:**

* Q1: Limiting the number of classes to four plus the correct class seems very constraining. Why did the authors choose this setting?

* Q2: How would concept annotators that also include distinguishing similar classes compare? The prompts (Appendix B) don’t focus on this aspect and therefore the models might yield annotations that can be confused with other classes.

* Q3: Regarding Fig. 3: What makes post-hoc textual annotations so much stronger than visual-grounded annotations?

* Q4: How do models compare on ImageNet in Tab. 3?

---

> ### Author Response · Authors · 2025-11-20
>
> ## W1: Is LLM-based class inference reliable?
> We’d like to clarify that our goal in this paper is not to guarantee that the LLM can reliably infer the correct class from its annotations. In fact, our key point is the opposite: relying on an LLM’s automatically generated concept annotations as “ground truth” for class inference in Concept‑Based Model training, as done in prior work [1–3], can be risky and potentially unreliable. Our experiments show that even when the LLM is guided through our full five‑stage “slow” refinement process—designed to make its concept reasoning explicit—it can still struggle to externalize its implicit knowledge in a fully accurate way.
>
> ## W2: Try top-5 concepts?
> See Appdx E.2
>
> ## W3: Lack of novel insight?
> We appreciate your observation that indeed it is widely recognized that current multimodal models do not perform well in capturing fine‑grained details. Our work aims to go one step further by providing a more nuanced analysis through the proposed FSE framework. As discussed around Line 425, our results suggest that such fine‑grained details may in fact be internally stored within the LLMs (as evidenced by the “fast mode” outcomes), but the community currently lacks effective mechanisms to extract these details from the black‑box in a transparent and explainable manner  (as evidenced by the “slow mode” outcomes). We view this work as an initial step towards addressing that gap, and we hope it can stimulate further research efforts in developing methods to make these latent fine‑grained representations accessible and interpretable.
>
> ## W4: Enhancing overall clarity
> We thank the reviewer for the feedback and will work on improving the overall clarity of the writing in the revised manuscript.
>
> ## C1: Utility-as-proxy
> The term "utility-as-proxy assumption" is our contribution to synthesizing and clarifying the concept that existing works implicitly use utility assessments to evaluate concept validity, as demonstrated in [4] and [5]. For example, [4] conducts a utility evaluation by randomly removing or adding concepts to test whether they affect the final class prediction. Similarly, [5] highlights the effectiveness of the proposed Vision-to-Concept Tokenizer by comparing the final classification accuracy. And we will refine the explanation of the “utility-as-proxy assumption” for improved clarity in the revised manuscript.
>
> ## Q1: Why four distractor classes?
> During our initial setup, we did not observe a substantial impact on final accuracy when varying the number of distractor classes. Therefore, we chose it to be 4 as a reasonable number of options, to strike a balance between providing a meaningful challenge for the LLM and maintaining manageable complexity.
>
> ## Q2: Would prompting to distinguish similar classes help?
> Our primary goal in this work is to evaluate the capabilities of LLMs in producing valid concept-class relations without prior exposure to the exact distinguishing features. Introducing explicit cues that directly differentiate similar classes in the prompt would, in effect, pre‑reveal the key information we aim to assess, rendering the evaluation less meaningful. Pre‑inserting those distinctive cues, however, would short‑circuit the evaluation and prevent us from truly measuring the LLM’s capability.
>
> ## Q3：Why textual is better then visual?
> This performance gap is largely stems from the different scenario setups inherent to post‑hoc textual versus visual‑grounded annotations. As described in Lines 131–147, post‑hoc annotations are generated at the class level: when queried, the LLM is already given the specific target class and can therefore draw upon its stored knowledge to deliver highly discriminative concepts relevant to that class. In contrast, visual‑grounded annotations are produced at the image level: the LLM is not told the class in advance. This inherently makes the task more challenging, and thus it is expected that post‑hoc performance would be higher than that of visual‑grounded annotations.
>
> ## Q4: ImageNet Results in Table 3?
> See Appdx E.1
>
> [1] Label-free concept bottleneck models
>
> [2] Language in a Bottle: Language Model Guided Concept Bottlenecks for Interpretable Image Classification
>
> [3] VLG-CBM: Training Concept Bottleneck Models with Vision-Language Guidance
>
> [4] Editable Concept Bottleneck Models
>
> [5] V2C-CBM: Building Concept Bottlenecks with Vision-to-Concept Tokenizer

---

### Official Review · Reviewer_eh4o · 2025-10-27

**Soundness:** 3
**Presentation:** 3
**Contribution:** 3
**Rating:** 6
**Confidence:** 3

**Summary:**

This paper investigates whether large language models (LLMs) and vision-language models (VLMs) can serve as reliable automated annotators for concept-based explainable AI (XAI) systems.
The work introduces the Fast and Slow Effect (FSE) framework, which evaluates the sufficiency of concept-class annotations without human supervision.
FSE models the annotation process as a continuum from a fast (opaque, intuitive) mode to a slow (explicit, conceptual reasoning) mode, and introduces a new metric, the Class Representation Index (CRI), to quantify whether generated concepts adequately represent the target class.
Through experiments on fine-grained and general vision datasets (e.g., CUB-200, Cars-196, CIFAR-100), the authors find that current LLM-generated annotations are often insufficient, particularly in fine-grained classification.
Surprisingly, “slow” conceptual reasoning frequently underperforms the “fast” intuitive mode, suggesting that models possess implicit visual knowledge they struggle to express explicitly.
The paper also critiques the “utility-as-proxy” assumption, that higher downstream accuracy implies better annotation quality, showing this correlation breaks down under FSE analysis.

**Strengths:**

- The FSE framework is an original and well-motivated contribution that systematically examines annotation sufficiency, filling a notable gap in XAI evaluation methods.

- The paper offers a convincing empirical analysis revealing a consistent gap between intuitive inference and explicit conceptual reasoning, raising important questions about the interpretability of LLM-driven explanations.

- The paper is well-organized, clearly written, and transparent about procedures, datasets, and metrics, which enhances reproducibility and conceptual clarity.

**Weaknesses:**

- The definition of sufficiency is somewhat circular and lacks grounding in formal interpretability theory. It might mistake a good result for true understanding.

- The five-stage annotation process (Background, Superclass, Salient, Detailed, Auxiliary) may heavily influence results. The robustness of findings to prompt variations or alternative hierarchies is unexplored.

- Although the paper critiques “utility-as-proxy,” its proposed CRI metric is still accuracy-based, measuring recoverability of class labels rather than interpretive richness or human-understandable sufficiency.

- The current research strictly limits its focus to image data. Therefore, its effectiveness is uncertain for other forms, such as text, tabular data, time series, or mixed-media reasoning.

**Questions:**

- How can CRI or FSE be extended to capture human interpretability, not just internal consistency?

- Would the same phenomena occur in text-only domains (e.g., sentiment analysis explanations) or multimodal reasoning tasks beyond classification?

- Can fine-tuning LLMs explicitly for conceptual disentanglement improve slow-mode sufficiency, or is this limitation architectural?

- Is there empirical evidence that higher CRI correlates with human trust or usability of explanations? Without this, the framework’s practical XAI relevance remains uncertain.

---

> ### Author Response · Authors · 2025-11-20
>
> ## W1: Informal definition?
> We thank the reviewer for raising this important concern about the definition of sufficiency. Our motivation for introducing sufficiency is closely tied to the main contribution of the paper: existing works on interpretability in LLM/VLM-generated concepts largely lack a principled way to supervise and validate those concepts from a theoretical perspective. Many prior XAI approaches [1–3] treat the annotations generated by LLMs/VLMs as essentially correct without formal validation — for instance, [1] explicitly assumes that “the LLM possesses a good amount of domain knowledge” without testing this assumption. In contrast, our notion of sufficiency views the full annotation process as defining a relation between concepts and classes (concept–class mapping). We propose to test whether this relation holds without requiring any external information or contextual cues (e.g., visual signals or other textual cues). This aims to formally capture whether the generated concepts contain enough information to consistently support the target classification task. We acknowledge that our current definition could be clarified to avoid potential circularity. In the revised manuscript, we will provide clearer definition.
>
> ## W2: Why the five-stage process?
> Our prompt design approach is intended to reflect and build upon existing methodologies in the field. Previous works [1] use three tiers (background, superclass, features), while [2,3] distinguish perceptual and descriptive levels. We combine these perspectives into Background, Superclass, Salient Features, Detailed Features, and Auxiliary Features. This extends prior tiers, integrates perceptual/descriptive elements, and adds Auxiliary Features for completeness. By positioning our framework in this context, we aim to underscore its value as a tool for critically assessing existing concept-extraction methods and identifying areas for improvement.
>
> ## W3: CRI VS utility-as-proxy
> CRI and utility-as-proxy differ fundamentally. Utility-as-proxy tests annotation validity by checking if combining concepts with imagery improves downstream accuracy; concepts are supplementary. CRI in our FSE framework uses only extracted textual concepts for classification. In Table 4, simulating utility-as-proxy with Fuse mode (Fast+Slow) yields CRI scores close to Fast alone and far above textual Slow mode.
>
>
> ## W4: More modalities
> We appreciate the reviewer’s comment and acknowledge that our current study is exploratory in nature. We will extend the framework to other modalities, such as text, tabular, time series, and mixed-modality settings, in future work.
>
> ## Q1: FSE beyond internal consistency
> We acknowledge that human trust and interpretability is a crucial aspect within the field of XAI. However, as outlined in Sections W1–W2, the primary objective of this work is to assess whether LLMs, when acting as annotators, can reliably understand and self-generate valid concept–class relations. Our focus is on evaluating the internal consistency of LLM-produced annotations, as this is a necessary foundation before pursuing more nuanced human-centric interpretability studies. In fact, the proposed FSE framework is not limited to internal consistency analysis—it can also facilitate investigations into human understandability, for example by leveraging chain-of-thought style reasoning traces from LLMs (see Appendix D). Nevertheless, we deliberately center our experiments and discussion on internal consistency, as ensuring reliability at this level is a prerequisite for confidently extending the approach toward richer human interpretability objectives in future work.
>
> ## Q2:  FSE in pure text contexts or other modalities
> In fact, the FSE framework also supports partially pure-text configurations. Please refer to Lines 131–147 and 249–264, where we describe the post-hoc textual annotation setting, as well as the experimental results in Figure 3, which corroborate the same finding. These demonstrate that FSE can operate effectively in text-only setups in addition to the multimodal cases presented. To further broaden its applicability, and as noted in Section W4, we plan to extend the framework to additional modalities—including text, tabular data, time-series, and mixed-modality scenarios—in future work.
>
> ## Q3：Fine-tuning?
> This work serves as an initial exploration. Considering the substantial time and computational cost of fine-tuning large language models, a rigorous and systematic evaluation might not be feasible within the rebuttal period. We sincerely appreciate your feedback and will address this in follow-up studies with more comprehensive experiments.
>
> ## Q4: Human trust?
> See Q1.
>
> [1] Label-free concept bottleneck models
>
> [2] Language in a Bottle: Language Model Guided Concept Bottlenecks for Interpretable Image Classification
>
> [3] VLG-CBM: Training Concept Bottleneck Models with Vision-Language Guidance

---

### Official Review · Reviewer_kQwZ · 2025-10-30

**Soundness:** 3
**Presentation:** 3
**Contribution:** 3
**Rating:** 6
**Confidence:** 3

**Summary:**

This paper investigates the problem of explainable AI in deep learning, whether large language models (LLMs) and vision-language models (VLMs) can generate reliable concept-based annotations. While previous studies have shown that LLMs can produce plausible annotations, the semantic adequacy of these annotations is unclear. To address this, the paper introduces the Fast and Slow Effect (FSE) framework, an autonomous self-evaluation methodology. FSE transitions annotation from a fast mode to a slow mode and evaluates each stage using a Class Representation Index (CRI) metric, which measures how well accumulated concepts represent their target classes. Experiments conducted on various visual classification datasets demonstrate that current annotations generated by LLMs lack sufficiently conceptual depth and need more effective and transparent strategies.

**Strengths:**

1. The paper is well written, and its motivation is clear.
2. This paper proposes a new explanatory framework that can automatically evaluate annotation sufficiency without human intervention.
3. The paper conducts experiments on fine-grained and common datasets with various model families and further re-examines the utility-as-proxy evaluation.

**Weaknesses:**

1. The paper initially discusses annotations for both LLMs and VLMs; however, the proposed methodology and experiments focus only on the explanation of vision tasks with VLMs, without evaluating textual explanation tasks or reporting results for LLMs.
2. The reliability of the concept-chain gathering process and the CRI metric as accurate measures of “semantic sufficiency” remains unclear, as they have not been rigorously validated through human evaluation.

**Questions:**

1. How sensitive is the prompt design to different vision tasks? The current five-stage refinement process is evaluated only on general-domain datasets. How well does this template perform in generating visual concepts for more specialized domains, such as medical imaging?
2. The concept generation process follows a hierarchical structure, for example, from background to detailed features. How does the order or number of steps in this concept chain influence the final results?

---

> ### Author Response · Authors · 2025-11-20
>
> ## W1: Focusing on VLMs only?
>
> Thank you for your constructive comment. Our intention with FSE is indeed to provide a unified evaluation framework that applies to both LLMs and VLMs. As described in Lines 131–147, we summarize the two mainstream paradigms in XAI: post-hoc textual annotation and visual-grounded annotation. These correspond respectively to the typical explanation behaviors of LLMs and VLMs. Regarding the experimental settings for both modes, they can be found in Figure 3 and Lines 418–429. We apologize for not making this clearer in the manuscript, and we will revise them in the revision stage.
>
>
> ## W2: Reliability and human study
> Quantitatively, the randomness test in Figure 3 demonstrates that variance across runs is minimal — the error bars are negligibly small, indicating reliable and stable measurements. From a theoretical perspective, our framework is motivated by the “fast–slow thinking” principle: the “fast” stage is naturally more variable, while the “slow” stage yields more robust results. This performance gap is explicitly captured and quantified in our FSE framework (Eq. 3, Line 302). The visual markers in Figure 3 (“star” vs. “dot”) once confirm this robustness difference clearly.
>
> Human study: We acknowledge that our current work does not include a large-scale human study. This is primarily due to the scale of annotations: for current LLM‑based XAI annotation pipelines, a single dataset often produces 3,000+ concepts [1–3], and in LaBo‑based pipelines this number can exceed 10,000 concepts. Performing a thorough human verification of such outputs would require far more time and resources than the rebuttal period allows.
> While it would be possible to conduct a simplified study by limiting validation to a small top‑k subset, such an approach risks overlooking a substantial portion of the explanatory content—especially in specialized domains—thereby compromising the validity and representativeness of the evaluation. Given the short rebuttal window and the effort required to design a scientifically sound protocol, we believe it is more appropriate to allocate sufficient time post‑rebuttal for a carefully planned human study. We agree that human validation is important and will pursue it in follow‑up work, employing targeted sampling strategies to balance feasibility and accuracy.
>
> ## Q1: Prompt sensitivity
> We examine prompt-design sensitivity across domains by distinguishing general-domain datasets—where classes are semantically diverse—from specialized-domain datasets—where classes are visually and semantically similar, requiring finer-grained concepts. Our five-stage refinement was evaluated on both types: CIFAR‑100 and Caltech‑101 (general) and fine-grained sets such as Car, Flower, and CUB‑Bird (specialized). Results in Table 3 and Figure 3 show that the framework adapts well: the “slow” LLM reasoning mode can outperform “fast” in certain general-domain cases, while fine-grained domains present different trade-offs (Lines 435–438). Medical imaging was excluded due to ethical and social-impact considerations and the absence of mature, open-source, medical-specific LLMs. We recognize its importance and plan future extension to this area, pending ethical review and resource availability.
>
>
> ## Q2: Will the number of steps affect the final results?
> Our concept generation process is intentionally designed to follow a coarse-to-fine hierarchy. Certain steps exhibit strong dependency relationships — for example, Step 4 (describing salient fine-grained features such as “red small wing” or “large black beak”) relies directly on the outputs of Step 3, which first establishes broader object parts (e.g., “wing,” “beak”). Rearranging such dependent steps breaks this logical progression and leads to incomplete or less coherent concept descriptions.
> We have experimented with reordering steps that have weaker dependencies. For instance, moving the “background” analysis from the beginning to the end produced minimal performance differences in both CRI scores and qualitative inspection. In summary, while the number of steps can be adjusted when certain layers are optional, the order is most influential when there are strong coarse-to-fine dependencies between steps.
>
> [1] Label-free concept bottleneck models
>
> [2] Language in a Bottle: Language Model Guided Concept Bottlenecks for Interpretable Image Classification
>
> [3] VLG-CBM: Training Concept Bottleneck Models with Vision-Language Guidance

---

### Note · Authors · 2026-01-26

I have read and agree with the venue's withdrawal policy on behalf of myself and my co-authors.

---

### Meta-Review · Area_Chair_Ynah · 2026-01-06

**Summary:**

The primary concerns center on the methodological reliability and lack of human grounding for the proposed evaluation framework. While the reviewers generally found the motivation and the "Fast and Slow Effect" (FSE) framework interesting, significant doubts were raised. Specifically, Reviewers eh4o and N1WD pointed out that using an LLM to evaluate whether another LLM's annotations are "sufficient" risks circularity and may simply measure internal consistency rather than true semantic depth or human-understandable interpretability. Additionally, the finding that LLMs struggle with fine-grained details was noted by Reviewer N1WD as a well-established observation, leading to questions regarding the paper's overall novelty. Finally, the absence of a human study to validate that the CRI actually correlates with human trust or understanding was a major weakness pointed out by multiple reviewers.

**Reviewer Concerns:**

The authors provided detailed responses that clarified several technical aspects of the work:

**Addressed**:
The authors successfully clarified the distinction between the CRI metric and the common "utility-as-proxy" assumption, explaining that CRI relies solely on textual concepts rather than supplementary visual imagery. They also provided a justification for their five-stage refinement process, grounding it in existing methodologies.

**Outstanding**: The most critical concern—the lack of human validation—remains outstanding. The authors argued that a large-scale human study was unfeasible during the rebuttal period due to the volume of annotations. While this is a practical constraint, it leaves the core metric (CRI) unvalidated as a tool for "Explainable" AI. Furthermore, the concern regarding circularity in the sufficiency definition was acknowledged by the authors but only promised as a future revision, leaving the current theoretical foundation of the framework shaky. The authors' defense of novelty—that the framework quantifies a "latent knowledge" gap—did not fully satisfy the concern that the core finding (LLM failure in fine-grained tasks) is already well-known.

**Reviewer Scores:**

Reviewer **kQwZ** (Initial Score 6): While the prompt sensitivity was addressed, the deferral of the human study to "follow-up work" leaves their primary concern about metric reliability unresolved.

Reviewer **eh4o** (Initial Score 6): The author's admission of "circularity" in the sufficiency definition and the acknowledgment that FSE primarily measures "internal consistency" over human interpretability likely weakens the initial rating.

Reviewer **N1WD** (Initial Score 2): The authors clarified that the "unreliability" of the LLM was actually the point of their study , which might slightly improve the score, but the reviewer's fundamental concerns about novelty and methodology remain.

---

### Decision · Program_Chairs · 2026-01-26

Reject